# Social Inequalities: Do They Matter in Asthma, Bronchitis, and Respiratory Symptoms in Children?

**DOI:** 10.3390/ijerph192215366

**Published:** 2022-11-21

**Authors:** Agata Wypych-Ślusarska, Karolina Krupa-Kotara, Ewa Niewiadomska

**Affiliations:** 1Department of Epidemiology, Faculty of Health Sciences in Bytom, Medical University of Silesia, 40-055 Katowice, Poland; 2Department of Biostatistics, Faculty of Health Sciences in Bytom, Medical University of Silesia, 40-055 Katowice, Poland

**Keywords:** health inequality, social determinants of health (SDH), bronchial asthma, respiratory symptoms, environmental factors, children

## Abstract

**Background:** Social inequalities (e.g., poverty and low level of education) generate inequalities in health. **Aim:** The aim of the study was to determine the relationships between indicators of social inequalities and the frequency of respiratory symptoms, asthma, and bronchitis in children. **Material and Methods:** In 2019, an epidemiological cross-sectional study on 3237 students from elementary schools in Silesia Voivodships (South Poland) was conducted. The students’ parents completed a questionnaire based on the International Study on Asthma and Allergies in Childhood (ISAAC). Social inequalities in the children’s families were determined according to parents’ education and professional status (working vs. unemployed), self-assessment of economic status, and housing conditions. To determine the influence of social factors on the occurrence of asthma, bronchitis, and respiratory symptoms, the odds ratio (OR) was calculated. **Results:** Children living in apartments with traces of mold had a higher risk of developing asthma (OR = 1.5, 95%CI: 1.17–1.96; *p* = 0.002) or bronchitis (OR = 1.4, 95%CI: 1.13–1.72; *p* = 0.002), wheezing attacks at nights (OR = 1.4; 95%CI: 1.01–1.93), wheezy in the last 12 months (OR = 1.6; 95%CI:1.24–2.08; *p* < 0.001), and chronic cough (OR = 1.9; 95%CI: 1.49–2.46; *p* < 0.001). Exposure to environmental tobacco smoke (ETS) was associated with higher risk of cough (OR = 1.5 95%CI: 1.22–1.96; *p* < 0.001) and dyspnea in the last 12 months (OR = 1.4; 95%CI: 1.04–2.00; *p* = 0.02). Low socioeconomic status (SES) was associated with increased risk of chronic cough (OR = 1.5; 95%CI: 1.09–2.03; *p* = 0.009) and increased risk of wheezy in the last 12 months (OR = 1.4; 95%CI: 1.06–1.97; *p* = 0.008). Asthma and bronchitis were not dependent on parents’ education or professional status. **Conclusions:** Social inequalities have significant impacts on the occurrence of respiratory symptoms, bronchitis, and asthma in children. Interventions aimed at preventing bronchitis and childhood asthma should also focus on social health determinants.

## 1. Introduction

According to the World Health Organization (WHO), “health inequities are systematic differences in the health status of different population groups. These inequities have significant social and economic costs both to individuals and societies” [1].

Inequalities in health undoubtedly result from social inequalities, which are the effect of “belonging to different groups, or occupying different social positions” [2]. In other words, social inequality means inequality of access to or opportunities for socially valued goods, such as material goods, education, power, prestige, or health. An important influence on health inequality is the social determinants of health (SDH), which, as defined by the WHO, means “non-medical factors that influence health outcomes. They are the conditions in which people are born, grow, work, live, and age and the wider set of forces and systems shaping the conditions of daily life” [3]. Determinants of the health status of various populations are increasingly being analyzed in relation to social risk factors. This takes into account socioeconomic status (SES), the living environment, including housing conditions and exposure to air pollution, the social environment and its security, access to health care, educational and employment opportunities.

Asthma is a disease whose etiology includes social determinants in addition to genetic and environmental factors [4,5,6,7]. They are related to environmental factors and are most often analyzed together. It is estimated that asthma may affect about 10% of children in Poland, but its prevalence may vary depending on environmental or social factors [8]. The cohort study ECAP (Epidemiology of allergic diseases in Poland) confirms the incidence of this disease in the pediatric population aged 6–7 years at the level of 4.4% among children living in urban areas and 3.9% living in rural areas [9,10]. The variation in asthma prevalence in different populations may be due to SDH. Studies in the United States indicate that certain ethnic groups may be particularly vulnerable to asthma [11]. The prevalence of the disease among Puerto Rican children (19.2%) or non-Hispanic blacks (12.7%) is higher than among non-Hispanic whites (8.0%) [11]. The reasons for the observed inequities may be the result of the accumulation and interaction of individual exposure factors and those operating at the community level, such as employment opportunities, a sense of security, or accessibility to health care.

Among the well-studied risk factors for bronchial asthma are internal and external allergens, including environmental tobacco smoke (ETS), house dust mites, molds, animal allergens, or air pollutants [10]. These can be related to the economic situation of specific groups, as well as their education. Without access to quality education, there is a greater risk of unemployment or low-paying jobs, which will equally be associated with the ability to live in a physically and psychologically safe environment. At the same time, economic deprivation will also limit access to good-quality food and a qualitatively and quantitatively well-balanced diet. Findings from both expective and observational studies indicate an association between reduced maternal vitamin D intake during pregnancy and a higher risk of asthma or wheezing in children [12]. Poverty also limits access to specialized health care and thus may influence more advanced forms of the disease. In addition, deprived populations are at higher risk for other diseases of civilization, which will ultimately translate into the occurrence of health inequities. To the best of our knowledge, no studies have been conducted in Poland to date on the social inequalities in asthma. Most studies have focused on the socioeconomic and environmental determinants of this disease [8,13,14]. However, simple analyses of the socioeconomic and environmental determinants of bronchial asthma do not fully explain the significance of social inequalities in this disease, but only indicate its social variation. For a full understanding of health inequalities, including asthma inequalities, a sociological perspective should be adopted. It is not about health differences, which often result from genetic or physiological differences, but about health inequalities, which should be analyzed as a chain of events leading to a specific health effect. The said chain of events will consist of many environmental or systemic conditions. Social factors affect health inequalities in conjunction with other factors that occur in the environment, not independent of them. Therefore, in addition to the social determinants of asthma, bronchitis, and respiratory symptoms, we include environmental factors.

In our earlier work, we compared the prevalence of respiratory symptoms and allergies in a group of children with and without asthma but also pointed out the variation in the prevalence of this disease depending on environmental conditions [8]. However, we did not analyze socioeconomic factors, and these are mainly responsible for the occurrence of social inequalities. The social criterion is relevant here, as it affects the understanding of the importance of social positions in health inequalities. Thus, in the present study, we focus on asthma inequalities, whereas previous analyses pointed to environmental asthma differences. This made it possible to highlight only environmental differences in the prevalence of respiratory symptoms, asthma, and bronchitis without answering the question of what generates these differences. We focused only on standard risk factors, but in understanding the importance of social inequalities in the prevalence of asthma, it is not enough to focus solely on the basic risk factors. According to the concept of causes of causes, it is necessary to consider what circumstances or conditions enable the known determinants of disease [15].

In this context, it seems important to determine the relationships between social inequalities and the frequency of respiratory symptoms, asthma, and bronchitis in children. By modelling the relationships between social and environmental indicators, we can determine whether the frequency of asthma, bronchitis, and respiratory symptoms depend on these factors. In short, the aim of the study was to determine the relationships between indicators of social inequalities and the frequency of respiratory symptoms, asthma, and bronchitis in children.

## 2. Materials and Methods

### 2.1. Characteristics of the Study Group

A cross-sectional study was conducted on a group of 3237 children attending elementary schools in the Silesian province (Poland) between 2018 and 2019 (Figure 1). The Silesian Voivodeship is located in the southern part of Poland and has the highest degree of urbanization and population density. The main branches of the go-economy are services and industry: coal mines, steel mills, and power plants are located here. At the end of 2020, the registered unemployment rate was 4.8% [16]. The Silesian Voivodeship has one of the highest annual emissions of particulate matter in Poland [17].

The study group included 1665 girls (51.4%) and 1572 boys (48.6%) aged 7–16 years. The average age of the subjects was 10.6 ± 2.2 years. The vast majority of the subjects were urban residents (2962, 91.5%); rural residents accounted for only 8.5% of the subjects (n = 275). Bronchial asthma affected 298 subjects (9.2%).

### 2.2. Eligibility Criteria

The study used cluster sampling: localities from each county of the Silesian province were randomly selected, and school principals from the selected localities were invited to participate in the study. The basic criterion for inclusion was the consent of the children’s parents to participate in the study, expressed by filling out a questionnaire. Participation in the study was anonymous and completely voluntary. The study complied with the provisions of the Declaration of Helsinki. The study was approved by the Bioethics Committee of the Silesian Medical University in Katowice (ID: PCN/CBN/0052/KB/190/22). In addition, the data collected were based on an anonymous questionnaire, to which parents gave voluntary written consent. The paper questionnaires were distributed face to face to the parents of school-aged children, and the absence of a completed questionnaire meant a lack of consent to participate in the study.

### 2.3. Research Tool

The research tool was an anonymous questionnaire based on the International Study of Asthma and Allergies in Childhood (ISAAC). Bronchial asthma was defined by a positive answer to the question “has a doctor ever diagnosed a child with asthma?” and the same question was asked to assess the prevalence of bronchitis. Among respiratory symptoms, wheezing in the last 12 months, night wake-up due to wheezing in the last 12 months, dyspnea in the last 12 months, and chronic cough were considered.

The indicators of social inequalities in the children’s families were mother’s and father’s education, parents’ professional status (working vs. unemployed), self-assessment of economic status, environmental tobacco smoke (ETS), presence of mold in homes, and type of heating (coal vs. central). The sociodemographic characteristic was measured objectively as parents’ education level, and socioeconomic status was measured as parents’ satisfaction with their employment status and income level.

All information on the children’s residential environment was obtained from the questionnaire responses. In addition, families’ socioeconomic status (SES) was rated as high, middle, or low as follows based on respondents’ subjective assessments:High SES—higher or secondary education, both parents working, and good self-assessment of economic status;Middle SES—middle or lower than the middle level of education, at least one parent working, and average self-assessment of economic status;Low SES—primary or vocational level of education, unemployed, and low self-assessment of economic status

The questionnaire was completed by the parents of the study children. All data were coded with appropriate symbols preventing the identification of patients by the Act of 29 August 1997 on the Protection of Personal Data (Journal of Laws 1997 No. 133 item 883).

### 2.4. Statistical Analyses

Statistical calculations were performed using the STATISTICA 13.0 program, Stat Soft Poland (StatSoft Poland sp. z o.o, Krakow, Poland). Measurable data were characterized using point series, and for nonmeasurable data, count tables and multivariate tables were used. To assess the relationships between qualitative variables, the chi-square test was used.

The influence of social risk factors on the occurrence of asthma, bronchitis, and respiratory symptoms was verified by multiple logistic regression with the occurrence of respiratory symptoms as the dependent variable. Adjusted models (adjusted odds ratios) with independent variables of sex or/and age, the presence of mold or dampness in dwelling, exposure to tobacco smoke, heating method, self-assessment of economic status, SES level, and parents’ education and professional status were used. The rationale for the selection of all independent variables was based on the chi-square test results.

We checked the collinearity of independent variables used in the model, and we did not identify autocorrelation between them. Statistical significance was determined at *p* < 0.05.

## 3. Results

### 3.1. Descriptive Analysis

Bronchial asthma affected 298 children (9.2%) and bronchitis 530 (16.4%). The most common respiratory symptoms were chronic cough (n = 306; 9.4%) and wheezy in the last 12 months (n = 298; 9.2%). Awakenings at night caused by wheezing attacks in the last 12 months affected 186 children (5.7%), and dyspnea in the last 12 months affected 153 (4.7%).

Most of the parents surveyed had a high school education (mothers—38.6%, fathers—36.6%) or higher education (mothers—45.0%, fathers—33.4%). The vast majority of parents of the children surveyed were employed; 33.4% of mothers and 10.0% of fathers reported being unemployed and thus lacking a source of steady income. Respondents most often described their economic situation as good (63.5%), and similar results were obtained for the SES: 66.3% of children came from families with high SES. ETS affected 37.2% of children, and 14.1% of respondents’ homes were heated with coal, while the presence of mold in apartments was reported by 24.3% of respondents.

Table 1 shows the prevalence of asthma, bronchitis, and respiratory symptoms according to social and demographic variables.

Boys were statistically significantly more likely than girls to have asthma, and bronchitis and were more often affected by attacks of shortness of breath and wheezing occurring in the past 12 months. The father’s and mother’s education differentiated the frequency of chronic cough. This symptom was statistically significantly more frequent in children whose parents had low education. A similar situation was observed concerning the activity of the children’s parents. Unemployment was associated with a higher incidence of chronic cough. Children exposed to ETS were more likely to have bronchitis and a higher prevalence of chronic cough and dyspnea in the last 12 months was observed. The analysis also showed that there was a correlation between coal heat and the presence of mold in apartments and a higher incidence of bronchial asthma in children. The presence of mold in apartments also showed a statistically significantly higher frequency of bronchitis, wheezing attacks at night, wheezy in the last 12 months, and chronic cough. Analysis including SES of families revealed correlations between low SES and frequency of wheezy in the last 12 months and chronic cough. The latter symptom was also statistically more frequent in children whose parents rated their economic situation as poor.

### 3.2. Multivariate Logistic Analysis

Concerning the socioeconomic factors for which statistical significance was observed between the analyzed groups, a multiple regression analysis was performed (Table 2).

A regression analysis (Table 2) confirmed that the presence of mold in the apartments increased the risk of asthma (OR = 1.5, 95%CI: 1.17–1.96), bronchitis (OR = 1.4; 95%CI: 1.13–1.72), wheezing attacks at nights (OR = 1.4; 95%CI: 1.01–1.93), wheezy in the last 12 months (OR = 1.6; 95%CI: 1.24–2.08) and chronic cough (OR = 1.9; 95CI: 1.49–2.46). ETS was associated with a higher risk of dyspnea in the last 12 months (OR = 1.4; 95%CI: 1.04–2.0) and chronic cough (OR = 1.5; 95%CI: 1.22–1.96), heating with coal increased the risk of asthma (OR = 1.4; 95%CI: 1.03–1.92). Poor economic status was observed to be associated with a higher risk of a chronic cough. Low SES was associated with chronic cough (OR = 1.5; 95%CI: 1.09–2.03) and wheezy symptoms in the last 12 months (OR = 1.4; 95%CI: 1.06–1.97). In the case of parents’ educational attainment and professional activity, it was noted that employment and higher education was protective with regard to chronic cough.

## 4. Discussion

### 4.1. This Work

Health inequalities are a topic that has been a permanent fixture in the scientific literature on public health, health sciences, and medical science more broadly. Ongoing research in this area addresses various aspects of population health, explaining the social and structural causes of variations in the health profiles of different groups. Additionally, concerning chronic diseases, their determinants in SDH are sought. This is because the latter affects differences in the prevalence and risk of developing a particular disease, thus leading to health inequalities.

Childhood bronchial asthma is increasingly becoming an interdisciplinary disease, in the sense that it is of interest not only to clinicians, pulmonologists, and allergologists but also to epidemiologists, including social epidemiologists. Due to its complex etiology, attention is increasingly focused on the socioeconomic determinants of this disease.

The purpose of this study was to try to answer the question of whether the social gradient and differences in socioeconomic situation, which are responsible for health inequalities, can affect the prevalence of asthma, bronchitis, and respiratory symptoms. The results of the survey, however, do not reveal a clear social gradient of the study group. Most of the respondents were people with a secondary or higher level of education, working professionally, and assessing their economic situation as good. Moreover, the SES scale showed that most of the respondents were of high SES. Thus, it can be observed that high SES was overrepresented in this study group and that we might consequently be lacking responses from some units in the target population. However, this type of problem is common not only in surveys but also in response to preventive measures or population-based screening programs [18,19,20]. On the other hand, taking into account environmental conditions, one can see greater diversity in the study population. Almost 40% of the children studied were exposed to ETS, almost every fourth apartment had traces of mold, and almost every fifth house was heated with coal. At the same time, the picture clearly presents an interpenetration of socioeconomic and environmental conditions, which should be considered inseparably in relation to analyses related to health inequalities. The results of our study also indicated that younger children were significantly more likely to have bronchitis, wheezing attacks at nights, wheezy in the last 12 months, and chronic cough. Trend analysis of the age of bronchial asthma diagnosis in Ka-nada children showed that the mean age of asthma diagnosis decreased from 4.7 ± 1.5 years in birth year 1993 to 2.6 ± 2.0 years in birth year 2000 [21]. The higher prevalence of respiratory symptoms in younger children may therefore be correlated with the diagnosis of asthma in the first years of life.

Studies show that racial and economic inequalities in asthma also include environmental risk factors [11,22,23,24]. Ethnicity is strongly correlated with poverty, and the two factors combined can amplify their effects and lead to social and economic deprivation [11]. In the case of asthma, this can include not only limited access to specialized care and opportunities for early detection, treatment, and control of asthma but also greater exposure to environmental or lifestyle factors. In the present study, due to the homogeneous structure of the population in Poland, analyses were not conducted by ethnic group or race. However, simple correlation analyses already showed that poor economic status and low SES were associated with a higher prevalence of the symptom of chronic cough lasting more than 3 months excluding the cold period. This symptom may not always be a symptom of asthma, but it certainly indicates respiratory problems and undoubtedly affects the patient’s quality of life [25,26]. In addition, chronic cough was associated with the lack of regular parental employment, the presence of mold in the apartments, and exposure to ETS.

Poor self-assessment of economic situation was associated with the risk of chronic cough, while low SES increased the risk of wheezy in the last 12 months. However, neither self-assessment of the economic situation nor SES level influenced the incidence of childhood bronchial asthma. This finding raised the question, however, of whether the disease is underdiagnosed in lower SES groups and overdiagnosed in better-off groups. Admittedly, this study analyzed respiratory symptoms that are associated with bronchial asthma and have occurred in the past 12 months. However, symptom alone is not necessarily a determinant of the disease [27], although a cohort study conducted in Tucson provides evidence that bronchial asthma in adults was closely associated with persistent wheezing symptoms in childhood [28]. Other studies have shown that low SES is a predictor of respiratory symptoms and is also associated with poorer asthma control [29]. Similarly, an analysis of population-based data from the United Kingdom proved that the higher the deprivation, the more severe the asthma symptoms, the later the diagnosis, and the higher the risk of hospitalization [22].

The higher prevalence of chronic cough and wheezy in the last 12 months in families with low economic status or low SES may be related to environmental exposures. Studies suggest that families with low SES are more exposed to such factors as indoor allergens, poor housing, and psychosocial stressors resulting from social marginalization, community, or family violence, among others [12,22,23,30]. The analyses conducted in our study showed that the most common determinant of asthma, bronchitis, wheezing attacks at the night, wheezy in the last 12 months and chronic cough was the presence of mold in children’s apartments. The negative effect of mold exposure on the risk of developing bronchial asthma is known in the literature [31]. This is confirmed, among other things, by the already historic Kohorot study, in which mold was found to be an independent risk factor for the development of asthma [32]. A recent study of children in an urban environment in Thailand also confirms that living in houses with mold is associated with a risk of wheezing and bronchial asthma [33].

Another confirmed risk factor for childhood asthma is ETS exposure. Although our own study did not reveal such an association, a correlation between ETS exposure and dyspnea in the last 12 months and chronic cough was observed in other works. The results of other studies, as well as meta-analyses and systematic reviews, clearly indicate that ETS exposure is an important risk factor for asthma and respiratory symptoms [34,35,36,37]. In addition to psychosocial factors, ETS exposure significantly influenced respiratory symptoms in a population living in Australia [37]. A meta-analysis of studies conducted on the Asian population yields similar conclusions. It unequivocally shows that ETS exposure is a significant risk factor for bronchial asthma [38].

Admittedly, both mold exposure and ETS can be categorized as environmental risk factors. However, as noted earlier, there is a close relationship between social and environmental risk factors. Some studies indicate that SES is subject to multiple pollutants that include environmental risk factors [29,39]. Residential exposures, smoking, ETS exposure, dietary habits, and obesity are among the modifiable environmental risk factors for respiratory diseases, including asthma, and they may affect the development of the disease regardless of the subjects’ SES. However, it is also emphasized that the modification of these factors is biased, attributed mainly to better-off populations [20]. People with lower SES often do not have the opportunity, resources, or knowledge that lead to the modification of environmental risk factors. Thus, this can be said to form a vicious circle of environmental and socioeconomic exposures. This in turn causes disparities based on the health profiles of the different groups, thus making health inequalities more visible. In conclusion, do health inequalities matter in the prevalence of asthma, bronchitis, and respiratory symptoms? Taking into account the environmental determinants of health, the analysis of the results of our own study confirms this regularity, despite the lack of a clearly differentiated social gradient of the population studied.

### 4.2. Strengths and Limitations

The strength of the presented survey is the representative sample group. In addition, the survey was conducted traditionally using a paper survey with parents of the children in the study population. This procedure avoided the common phenomenon of fake responders that characterizes online surveys shared. The survey was a standardized questionnaire, which also had a significant impact on the quality of the data presented and the possibility of comparing it with other ISAAC studies. An epidemiological cross-sectional study was performed using the questionnaire used in the International Study of Asthma and Allergies in Childhood (ISAAC). It is a standardized tool, validated and adapted to the cultural and social context of each country, that allows for a reliable estimation of the prevalence of respiratory diseases and symptoms, especially of childhood bronchial asthma.

The study presented here made it possible to identify sensitive risk factors and populations that are particularly likely to be affected by respiratory diseases such as asthma and bronchitis as well as respiratory symptoms. This is particularly relevant in the era of the ever-present problem of COVID-19. In addition, the study was conducted at a time before the pandemic, which means that asthma and reported respiratory symptoms were not due to current SARS-CoV-2 virus infection.

The main limitations of the study are due to the study model adopted: a cross-sectional study, which makes it impossible to draw causal conclusions. Rather, the results presented are of statistical association, so we do not show that a change in the frequency of exposure in a population will result in a change in the frequency of the outcome. This fact, however, does not detract from the significance of the study conducted but sensitizes us to the proper interpretation of the data and conclusions. Another limitation of a cross-sectional study is incomplete information on exposure. Exposure and effect data are collected at a single point in time, and even the best-constructed questionnaires will not indicate with measured accuracy the dose of exposure and its duration. Therefore, it is important in this case to use a standardized survey instrument that has been validated in many other studies and applied to populations in other countries. In our case, it was the standardized ISAAC questionnaire. Various methods are often used in questionnaire studies to assess ETS exposure: the number of cigarettes exposed per day; the number of persons smoking in the house; exposure during a mother’s pregnancy; or the presence of a father, mother, or any family member who is a smoker [37]. Regardless of the definition of exposure adopted, the results clearly indicate the potentially harmful effects of ETS. In studies of health inequalities, there is no single criterion against which these inequalities should be measured. Inequalities can be related to the incidence of disease, hospitalization, disease exacerbations, or mortality [39]. There is a similar problem in the determination of SES. Here, too, there is no clear criterion [40,41].

### 4.3. Future Work

Our attempt to determine the SES of families highlighted the importance of this factor concerning the symptom of chronic cough. However, a similar relationship did not apply to asthma. Notwithstanding that finding, it is not excluded that this is a result of the small social gradation of the study group, and future studies should also focus on methods of reaching groups of respondents who would be representative of low SES groups. Due to this fact, our study cannot be considered representative, and the conclusions made apply only to the population relevant to the analyses conducted. Future research on the significance of social inequalities in the prevalence of respiratory diseases, especially asthma, should take into account broad macro-social and macroeconomic conditions. The economic crisis, rising inflation, and the stagnation of wages, as well as the increasing prices of products and services in Poland, may worsen the economic situations of many families, including their possibilities for diagnosis or treatment. The current political situation in Europe in connection with the ongoing war in Ukraine and the energy crisis affecting Polish families are also not without significance. It manifests itself, among other things, in problems with the supply of coal, the rising prices of this resource, or the need to purchase coal of lower quality. In addition, the Polish parliament repealed the quality standards for solid fuels for two years and allowed brown coal, which has meant a deterioration in air quality, which in Poland has for decades been among the most polluted in the entire European Union. With regard to the Silesian Voivodeship, which is characterized by the highest population density in Poland and significant urbanization, this situation may have a particularly negative impact on the state of the respiratory tract in particular populations, especially children and the elderly. In this context, it seems crucial to focus attention not only on the population of all school-aged children, both healthy and those diagnosed with bronchial asthma, but additionally on sick children only, in order to analyze the impact of social inequalities on the course of the disease and social functioning, both of the children themselves and of their families.

In addition, it is also worth considering psychosocial factors that may trigger asthma in predisposed individuals or exacerbate its symptoms in patients. Undoubtedly, attention to residential neighborhood attributes can enrich such analyses and bring us closer to understanding the importance of social inequalities in the prevalence of bronchial asthma. Studies focusing on various aspects of social inequality indicate that neighborhood income inequality significantly affects other determinants of children’s social functioning, such as school attendance [42]. The latter characteristic undoubtedly affects the ability to obtain an adequate level of education, which affects employment opportunities and, ultimately, an individual’s economic position. The links between economic situation and bronchial asthma have already been explained in this paper. Unsafe and poor neighborhoods can also affect the occurrence of chronic stress and strong emotions, which in turn can contribute to the exacerbation of asthma symptoms. Therefore, for a full understanding and explanation of inequalities in asthma, residential neighborhood attributes should be included in future studies.

## 5. Conclusions

Low socioeconomic status, both in the self-assessment of the economic situation and in the developed scale, determines the frequency of chronic cough and wheezy in the last 12 months. Parental education does not affect the frequency of respiratory symptoms, asthma, or bronchitis. The observed association of environmental factors (mold, ETS, and coal heating) with the prevalence of respiratory symptoms and asthma may be due to the social situation of the subjects, but further in-depth studies are needed to confirm or exclude this relationship. The identified risk factors may improve the understanding of the social etiology of asthma and the importance of social inequalities in the prevalence of the disease and respiratory symptoms.

## Figures and Tables

**Figure 1 ijerph-19-15366-f001:**
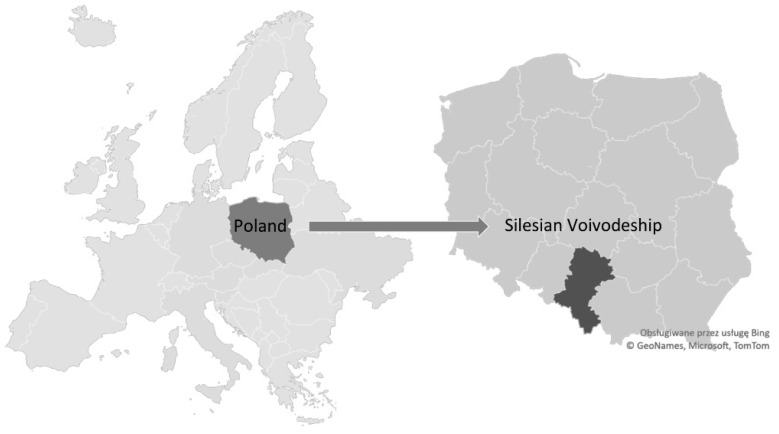
The place of the survey (map developed using MS Excel 2019).

**Table 1 ijerph-19-15366-t001:** Presence of asthma, bronchitis, and respiratory symptoms in children (7–16 years old) according to the socioeconomic and demographic profiles of their parents.

Social and Demographic Variable	Total n (%)	Asthma n (%)	*p* *	Bronchitis n (%)	*p* *	Wheezing Attacks at Nights n (%)	*p* *	Wheezy in the Last 12 Months n (%)	*p* *	Chronic Cough n (%)	*p* *	Dyspnea in the Last 12 Months n (%)	*p* *
**Sex**													
male	1571 (49.0)	180 (11.4)	**<0.001**	288 (18.3)	**0.03**	102 (6.5)	0.07	165 (11.0)	**0.01**	159 (10.1)	0.2	90 (5.73)	**0.009**
female	1664 (54.0)	118 (7.1)		242 (14.5)		84 (5.0)		133 (8.0)		147 (8.8)		63 (3.79)	
**Mother’s education ****													
low ***	527 (16.4)	47 (8.9)	0.2	84 (16.0)	0.9	29 (5.5)	0.8	51 (10.0)	0.6	64 (12.1)	**0.006**	24 (4.5)	0.1
middle	1238 (38.6)	127 (10.3)		200 (16.1)		75 (6.1)		120 (10.0)		126 (10.2)		69 (5.6)	
high	1441 (45.0)	121 (8.4)		241 (16.7)		80 (5.6)		124 (9.0)		112 (7.8)		57 (4.0)	
**Father’s education ****													
low	901 (30.0)	88 (9.8)	0.4	150 (16.6)	0.9	60 (6.7)	0.2	92 (10.0)	0.1	101 (11.2)	**0.01**	50 (5.5)	0.3
middle	1100 (36.6)	104 (9.4)		175 (15.9)		53 (4.8)		86 (8.0)		93 (8.4)		46 (4.2)	
high	1005 (33.4)	81 (8.0)		165 (16.4)		60 (6.0)		98 (10.0)		75 (7.5)		43 (4.3)	
**Mother’s work activity**													
working	2479 (76.6)	219 (8.8	0.2	393 (15.8)	0.1	138 (5.6)	0.4	220 (9.0)	0.2	215 (8.7)	**0.005**	113 (4.6)	0.5
non-working	755 (33.4)	79 (10.5)		137 (18.1)		48 (6.4)		78 (10.0)		91 (12.0)		49 (5.2)	
**Father’s work activity**													
working	2909 (90.0)	262 (9.0)	0.2	477 (16.4)	0.9	165 (5.7)	0.5	266 (9.1)	0.6	256 (8.8)	**<0.001**	131 (4.5)	0.06
non-working	325 (10.0)	36 (11.1)		53 (16.3)		21 (6.5)		62 (9.8)		50 (15.4)		22 (6.8)	
**ETS**													
yes	1202 (37.2)	125 (10.4)	0.7	217 (18.0)	**0.04**	72 (6.0)	0.6	118 (9.8)	0.4	143 (11.9)	**<0.001**	70 (5.8)	**0.02**
no	2029 (62.8)	173 (8.5)		312 (15.4)		114 (5.6)		180 (8.9)		163 (8.0)		83 (4.1)	
**Type of heating**													
central heating ****	2759 (85.9)	241 (8.7)	**0.02**	445 (16.1)	0.5	155 (5.6)	0.7	239 (8.7)	0.05	250 (9.1)	0.1	127 (4.6)	0.6
coal burning	453 (14.1)	54 (11.9)		79 (17.4)		27 (6.0)		52 (11.5)		52 (11.5)		23 (5.1)	
**Presence of mold**													
yes	782 (24.3)	94 (12.0)	**0.001**	156 (19.9)	**0.002**	56 (7.2)	0.05	97 (12.4)	**<0.001**	110 (14.0)	**<0.001**	45 (5.7)	0.1
no	2440 (75.7)	203 (8.3)		372 (15.2)		129 (5.3)		199 (8.2)		193 (7.9)		107 (4.4)	
**Economic situation (self-assessment)**													
good	1998 (63.5)	176 (8.8)	0.5	327 (16.4)	0.6	112 (5.6)	0.3	188 (9.4)	0.2	180 (9.0)	**0.01**	90 (4.5)	0.4
average	1066 (33.9)	106 (9.9)		176 (16.5)		60 (5.6)		89 (8.4)		104 (9.8)		52 (4.9)	
bad	82 (2.6)	8 (9.8)		10 (12.3)		8 (9.8)		11 (13.4)		15 (18.3)		6 (7.3)	
**SES**													
high	1932 (66.3)	167 (8.6)	0.2	303 (15.9)	0.5	105 (5.4)	0.5	171 (8.8)	0.05	168 (8.7)	**0.04**	90 (4.5)	0.5
middle	462 (15.9)	49 (10.6)		75 (16.2)		24 (5.2)		35 (7.6)		37 (8.0)		52 (5.0)	
low	518 (17.8)	55 (10.6)		94 (18.1)		35 (6.8)		61 (11.8)		62 (11.0)		6 (7.3)	
**Age [years]**	X ± SD	**Asthma** X ± SD	***p* ***	**Bronchitis** X ± SD	***p* ***	**Wheezing** **attacks at** **nights** X ± SD	***p* ***	**Wheezy in the last 12 months** X ± SD	***p* ***	**Chronic cough** X ± SD	***p* ***	**Dyspnea in the last 12 months** X ± SD	***p* ***
no	10.6 ± 2.2	10.6 ± 2.2	0.5	10.7 ± 2.2	**0.001**	10.6 ± 2.2	**0.002**	10.7 ± 2.2	0.2	10.7 ± 2.2	0.1	10.6 ± 2.2	0.2
yes **		10.6 ± 2.2		10.3 ± 2.1		10.1 ± 2.1		10.3 ± 2.2		10.3 ± 2.2		10.4 ± 2.2	

Data presented as numbers and percentages; * *p*-value for the chi-square test; Data presented as mean and standard deviation X ± SD; * *p*-value for the *t*-Student test; ** yes—presence of asthma, bronchitis, or respiratory symptoms; *** low—primary or vocational; **** central heating, gas heating; ETS—environmental tobacco smoke; SES—socioeconomic status.

**Table 2 ijerph-19-15366-t002:** Adjusted odds ratios (OR) and 95% confidence intervals related to determinants of observed respiratory symptoms.

Health Problem	Determinants of Asthma, Bronchitis, and Respiratory Symptoms (* Reference Group)	OR (95%CI) Sex- and/or Age-Adjusted	*p*-Value
Asthma	Sex (male/female *)	1.7 (1.33–2.16)	**<0.001**
	Presence of mould in apartments (yes/no *)	1.5 (1.17–1.96)	**0.002**
	Type of heating (coal/clean *)	1.4 (1.03–1.92)	**0.03**
Bronchitis	Sex (male/female *)	1.3 (1.10–1.59)	**0.003**
	Presence of mold in apartments (yes/no *)	1.4 (1.13–1.72)	**0.002**
	ETS (yes/no *)	1.2 (0.99–1.46)	0.05
	Age [years]	0.9 (1.09–1.59)	**0.001**
Wheezing attacks at nights	Presence of mold in apartments (yes/no *)	1.4 (1.01–1.93)	**0.04**
	Age [years]	0.9 (0.82–0.95)	**0.002**
Wheezy in the last 12 months	Sex (male/female *)	1.4 (1.06–1.71)	0.01
	Presence of mold in apartments (yes/no *)	1.6 (1.24–2.08)	<0.001
	SES (low/middle/high *)	0.9 (0.59–1.26)/ 1.4 (1.06–1.97)	0.08/ **0.008**
Chronic cough	Mother’s education (low */middle/high)	0.8 (0.58–1.11)/ 0.6 (0.41–0.79)	0.6/ **<0.001**
	Father’s education (low */middle/high)	0.7 (0.52–0.94)/ 0.6 (0.43–0.82)	0.5/ **0.02**
	Mother’s work activity (working/not working *)	0.7 (0.54–0.91)	**0.007**
	Father’s work activity (working/non-working *)	0.5 (0.37–0.71)	**<0.001**
	ETS (yes/no *)	1.5 (1.22–1.96)	**<0.001**
	Presence of mould (yes/no *)	1.9 (1.49–2.46)	**<0.001**
	Self-assessment of economic situation (bad */average/good)	0.4 (0.24–0.78) 0.5 (0.26–0.87)	**0.007/** 0.07
	SES (low/middle/high *)	0.9 (0.64–1.34)/ 1.5 (1.09–2.03)	0.14/ **0.009**
Dyspnea in the last 12 months	Sex (male/female *)	1.5 (1.11–2.14)	0.01
	ETS (yes/no *)	1.4 (1.04–2.00)	0.02

* reference group; OR (95%CI)—odds ratio (95% confidence interval); ETS—environmental tobacco smoke; SES—socioeconomic status.

## Data Availability

Not applicable.

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
