# Peer review of "Social Inequalities: Do They Matter in Asthma, Bronchitis, and Respiratory Symptoms in Children?"

_ijerph, 2022, doi:10.3390/ijerph192215366_

Round 1

Reviewer 1 Report

The topic is interesting for everyone to know whether social inequalities have something to do with child's respiratory health problems. However, the authors only conduct univariate logistic regression analysis to show preliminary results without further work to find out the most important factors that are associated with respiratory symptoms (hopefully SES) using multivariate regression analysis. For example, the results of univariate logistic regression analysis show 8 significant factors associated with chronic cough. The authors are supposed to conduct multivariate regression analysis to identify the truly significant factors, or in another way, to adjust for other factors while doing regression analysis for SES. Without this portion of work, this study is not considered complete.

Minor issues are listed as follows:

Full names of abbreviations must be given at the first time, such as ETS in Abstract, SES in Introduction.

The face-to-face survey method (Line 311) should be presented earlier in the section of Materials and Methods.     

Author Response

Dear Reviewer,
We sincerely thank you for taking the time to carefully review our manuscript and sending many valuable comments to improve it. We are enclosing your responses, we hope they will be sufficient and satisfactory to accept the manuscript for publication. 
With best regards, Authors

Reviewer 2 Report

The work presented is interesting, although the results and conclusions are easily predictable. The methodology is very simple. The authors should specify whether they have used logit or probit models. They should also provide the usual statistics in this methodology to see the goodness of fit, as well as the concordance percentages.

In the abstract the authors use two acronyms that should be avoided, as the reader does not know what they refer to: STD and SES. Both are described later in the text, but the abstract should be understandable without reading the rest of the manuscript.

The main problem with this manuscript is its lack of originality. Although it includes some new variables, it lacks novelty due to previous work published by the authors on this same topic and using the same sample. Indeed, the percentage of similarity shown by the TURNITIN program is high. Specifically, the previously published work which, moreover, the authors have taken care not to cite is the following:

* Wypych-Slusarska, A. et al. (2022). Respiratoty Symptons, Allegies, and Envrionmental Exposures in Children with and without Asthma. IJERPH. https://doi.org/10.3390/

ijerph191811180

Author Response

(The authors gave the same response as above.)

Reviewer 3 Report

Dear Authors; I found this work an interesting discussion on the impact of social inequalities on the Asthma, Bronchitis, and Respiratory Symptoms in Polish Children. As a veteran reviewer for the journal, I find its current status 4 degrees below journal publication standards. It needs serious extra work to arrive publication quality. Please make sure to secure enough time for the revision from the journal editorial staff.  Regards, P.S.

[1] Writing:

1-1 Abbreviations: add list of used abbreviations in the manuscript at end right before references.

1-2 "3. Results":  break it down to two subsections: 3.1. Descriptive Analysis; 3.2. Multivariate logistic analysis

1-3 "4.Discussion": break it down to 4 subsections: 4.1. This work  4.2. strengths and limitations 4.3. future work.  Move section "5.Strengths and Limitations" to 4.2. here. Add section "4.3.future work" then.

1-4 Compare your work in the Discussion Section in terms of consistency with studies considering SES determinants for standards EDI outcomes:

Some sources for citations and comparison:

Muhajarine, N.; McRae, D.; Soltanifar, M. Aboriginal Status and Neighbor hood Income Inequality Moderate the Relationship between School Absenteeism and Early Childhood Development.  Int. J. Environ. Res. Public Health 2019, 16, 1347. https://doi.org/10.3390/ijerph16081347

More studies are here: https://www.mdpi.com/journal/ijerph/special_issues/child_health

1-5 Add the Map plot of Poland and the Silesia Voivodships region to the paper. This is an international journal and your non-Eu readers should be able to know with minimum effort where the study was conducted. Your Map plot has two panels: (a) Poland; (b) Silesia Voivodships

Recommended Sources:

Poland:

https://en.wikipedia.org/wiki/Poland#/media/File:EU-Poland_(orthographic_projection).svg

Silesia Voivodships:

https://en.wikipedia.org/wiki/Silesian_Voivodeship#/media/File:Slaskie_(EE,E_NN,N).png

https://www.flaggenlexikon.de/flag-encyclopedia/poland_voivodeship.htm

[2] Statistical:

2-1 Add age covariate statistics in Table 1 and Table 2

2-2 Defend your choice of univariate logistic regression versus multivariate logistic regression. Note: Statistically, when each covariate is significant in a univariate logistic regression, statisticians consider considering fitting multivariate logistic regression model with several models.

2-3 Table 2: Each model in each line of the Table 2 must be having controlling variables such as "age" and "sex".  Rerun all lines with these two extra variables and report the new results. 

2-4 Missing Plots: Add some plots as the results of Multivariate logistic regression for Asthma and Bronchitis outcomes in terms of "age" each categorized in terms of their determinants in the Table.2- See Figure.5. in the following reference to get the ideas:

Muhajarine, N.; McRae, D.; Soltanifar, M. Aboriginal Status and Neighborhood Income Inequality Moderate the Relationship between School Absenteeism and Early Childhood Development. Int. J. Environ. Res. Public Health 2019, 16, 1347. https://doi.org/10.3390/ijerph16081347

Author Response

(The authors gave the same response as above.)

Reviewer 4 Report

The topic is relevant because it focuses on analysing health determinants in the most prevalent respiratory pathology in children, which has impact on public health policy. The study design is a cross-sectional study to identify for associations between socio-demographic factors and diagnosis of asthma, bronchitis or disclosure of respiratory symptoms. The manuscript is well written in a clear language and the methodology is well described. 

However, there are some issues that need to be addressed:

In the abstract, acronyms should be avoided, or at least properly referenced, e.g. ETS (line 22).

Introduction has a poor bibliographic support that must be improved. For example, in: "Asthma is a disease whose aetiology includes social determinants ..." (lines 50-51), only one meta-analysis on asthma risk in children born by caesarean section is cited, which does not focus on social inequalities. Please, provide several references of recent systematic reviews or meta-analyses, preferably.

When discussing the prevalence of asthma in children, it should be noted that the ECAP cohort study is focused on allergic asthma (lines 54-56). The authors might consider moving the citation to the next paragraph on the aetiology and intrinsic and extrinsic risk factors.

Material and methods:

The authors should provide a justification for the choice of the geographic location, as well as a socio-demographic description of the Silesian Voivodships region. This would help to interprete the results.

Although participants gave their informed consent and anonymity was guaranteed, the study must have been approved by a bioethics committee, as medical information and socio-demographic data on the children is collected. This is a mandatory requirement in any health research, including qualitative studies, and not exclusively for experimental designs. If this is a nested study, at least the approval of the ethics committee of the main study should be provided.

The authors should emphasize in the objectives that socio-demographic characteristics are based, for educational level, on objective data and for socio-status the parents' perception or satisfaction with their employment status and income level.

Results:

Line 15: Add "m" to "Most".

In the table:

-Detail the title, for example: "Presence .... symptoms in children (7-16 years old) according to socio-economic and demographic profile of their parents".

-Include the "age of children" as a independent variable distinguishing several ranges, for example, and comment these in both Results and Discussion sections.

-Justify the independent variables (sex, mother's education, etc.) to the left and highlight in italics or bold.

-Indicate with bold or asterisk the statistically significant values.

-Use the same categories indicated in the description of the Type of heating variable in the Methods section ("central heating" instead "clean", and "coal burning" instead "coal stove").

Discussion:

The discussion could be abbreviated.

The authors may contextualize the results according to the sociodemographic characteristics of the region described in the Methods section.

Please, discuss if age of children was a significan variable in this study and compare to other findings. 

Line 214: This is not a question, please, delete the question mark.

Lines 260-262: Please, correct "have to" by "have wondered", and remove the question mark.

Conclusions:

Line 360: What is "ba-da"? Please, clarify.

Bibliography

The authors must use the same citation system.

Author Response

(The authors gave the same response as above.)

Round 2

Reviewer 1 Report

Why only adjust for age and sex? For example of chronic cough, the authors should adjust for all other variables to identify whether SES is significantly related to the heath problem. If some of the variables (e.g., ETS, presence of mould) are highly correlated with SES, that would be an issue of collinearity.  In that case, you do not adjust for those variables that are highly correlated with SES. 

Correlation analysis (Spearman, I suppose) containing all variables should be conducted to find out the existence of collinearity. 

Author Response

First, we would like to thank for this analysis of the manuscript. We agree with the majority of the Reviewers’ comments. Our current replies are written in a blue. We made several changes in our manuscript believing they address all the Reviewers’ comments (also marked in red – earlier review).

In Table 2, the authors included the effect of each variable separately, among those that showed significant variation (Table 1). Thus, for chronic cough, the crude odds ratio was determined separately for each predictor, including SES, independently of the others.logit(p)=X1B1

Then, each predictor was taken into the model, as well as gender and age. This is how the adjusted odds ratio was determined.

logit(p)=X1B1+SexB2+AgeB3

Figure 2, on the other hand, already takes into account multivariate modeling, where a collinearity of less than 0.4 was taken into account before determining the model by backward stepwise regression.

Re: page 4, lines 172-173:

We checked the collinearity of independent variables used in the model, and we did not identify autocorrelation between.

Only for chronic cought the model (Figure 2) was changed due to the significant correlation of mother/father education; Self-assessment of economic situation and SES.

Page 7, line 241:

Only for chronic cough was a significantly lower risk confirmed for father's work activity higher maternal education.

Page 8 – Figure 2

Reviewer 2 Report

The work is interesting and, therefore, could be published in its present state. However, from my point of view, the novelty is still insufficient due to other previous studies by the authors. It is up to the editor to judge whether that novelty is sufficient for the journal.

Author Response

Dear Reviewer,

Thank you for your positive feedback and interest in our work. The topic addressed in this paper is similar to the previous paper (Wypych-Åšlusarska, A.; Grot, M.; KujawiÅ„ska, M.; Nigowski, M.; Krupa-Kotara, K.; Oleksiuk, K.; GÅ‚ogowska-Ligus, J.; Grajek, M. Respiratory Symptoms, Allergies, and Environmental Exposures in Children with and without Asthma. Int. J. Environ. Res. Public Health 2022, 19, 11180. https://doi.org/10.3390/ijerph191811180), but has important differences that we tried to point out in the introduction of this manuscript. The earlier paper focused only on environmental factors. In it, we did not consider the social gradient and did not examine whether it affects the prevalence of asthma, bronchitis and respiratory symptoms. When analyzing social inequalities in the prevalence of the aforementioned diseases and symptoms, environmental factors cannot be ignored - low educational level, lack of work, low self-esteem of economic situation or low SES can affect the living conditions of children's families. Therefore, these factors were again analyzed alongside social determinants. The present study, in our opinion, fits into the thematic scope of social epidemiology. We wanted to cover a broad spectrum of different determinants of health and thus approach the topic holistically. 

Best regards, Authors

Reviewer 3 Report

Dear Authors, I think my major concerns were addressed satisfactorily. Regards.

Author Response

Dear Reviewer, 
Thank you very much for reviewing our manuscript again. We appreciate your help and your time. We have made a few more improvements to our manuscript, highlighted in red. We hope they will further improve the quality of our study. 
With best regards, Authors

Reviewer 4 Report

After reviewing the authors' responses to comments and changes made to the manuscript, I agree that it be accepted for publication.

Author Response

(The authors gave the same response as above.)
